# Movement, Play, and Games—An Essay about Youth Sports and Its Benefits for Human Development

**DOI:** 10.3390/healthcare11040493

**Published:** 2023-02-08

**Authors:** Miguel Nery, Isabel Sequeira, Carlos Neto, António Rosado

**Affiliations:** 1Faculdade de Ciências Sociais e Tecnologia, Universidade Europeia, 1500-210 Lisboa, Portugal; 2Self-Psicologia e Psicoterapia, 1150-278 Lisboa, Portugal; 3Faculdade de Motricidade Humana, Universidade de Lisboa, 1649-004 Lisboa, Portugal

**Keywords:** youth health, movement, play, sports, body

## Abstract

The acknowledgment of the qualities and features of the world is made through the body, movement, and imagination. During their development, children learn new skills, complexify their thoughts, and become more autonomous. The progressive increase in motor repertoire in children reflects a more unified and solid self. Nowadays, there is a generalized restriction of the movement of children. It starts at home when parents establish rigid and/or phobic attachments with their children; it can be also observed at school which is more and more based on rigid learning rhythms and obsessive ideas about students’ performance, and finally in urban areas where free and outdoor play has considerably decreased during recent decades. The current lifestyles in Western societies resulted in a decrease in play among children. The culture influences the dominant types of psychopathology and, during childhood, mental suffering is often expressed with the increase (turmoil) or decrease (inhibition) of the body movement. Sports are underpinned by movement and play; they are a powerful tool in health promotion and an excellent way to assign meaning to movement. This work is an essay about the importance of play and youth sports in child development.

## 1. Introduction

This essay article addresses the importance of play and youth sports in child development. We focus mostly on early stages of sport engagement, when play should be the core activity of childhood. Different theoretical approaches are considered, including child motor behavior, sports sciences, psychology, and psychoanalysis. The manuscript is divided into three parts, and each one of them is divided into its main topics and sub-topics.

PART I is dedicated to child development, and aims to provide a general framework for the upcoming sections. We start with a brief review of early phases of child development (psychomotor development), before moving forward until the beginning of more structured sport practice. A special attention is provided to the body as a means of self-expression and communication with others, and to different types of play (and its development) during childhood. It includes topics such as body, attachment, motor behavior, and play.

PART II—entitled Actual Constraints on Child Development—focuses on recent issues that constrain child (natural) movement, and have negative impact on their development. Despite the impressive economic development in the last half century in Western societies, with considerable positive impact on child health and education, modern lifestyles face new obstacles. Among these is a severe decrease in physical activity and play among youngsters. We describe how child movement and (free unstructured) play have become less accepted in settings such as home, school, and street. Later, we focus on the analysis of problems that result from it. We do not intend to make an in-depth analysis of child psychopathology; the focus is on the difficulties mostly expressed through the body (although related to overall development). We divided such problems into opposite poles related to a lack or excess of movement.

Finally, in PART III, we focus on the role of sports in child development. Here, we analyze the symbolic meaning of sports, and how it may contribute to fostering healthy development. This final section aims to provide a reflection about the use of sports as a tool to tackle issues. To do so, we start with a brief description of three models that emphasize the role of sports in child development. Although being generally perceived as a social good, engagement in sports is not always positive, and can also become a source of suffering due to different types of abuse existent in this context. We briefly address some types of violence and abuse that may be found in sports, and the increasing importance of safeguarding. We finish with a reflection about the potential of sports as a tool to promote healthy development (when properly conducted).

## 2. PART I. Child Development. Play and Motor Behavior

### 2.1. Body

Play is a key element in child development. The body, movement, and the imagination are very important means for children to explore and progressively understand their environment. The body includes both “somatic” and “relational” parts, and these are interconnected. The somatic part of the body relates to the maturation process, musculoskeletal system, and physiology, and it is mainly approached by scientific areas such as biology, chemistry, and some disciplines of medicine such as neurology and pediatrics, among others. On the other hand, the relational part of the body relates to attachment (affect, emotions, and feelings) between the child and his/her significant others and environment, and it is mostly approached by scientific areas such as psychology, psychoanalysis, and social sciences.

The body must be understood based on an integrative perspective that considers both somatic and relational parts, their interconnection, and mutual influence. We consider the tangible part of the body, but with extensions that go beyond its physical limits, through the attachment and connections established during the personal history of the individual, and the development of his/her autobiographical memory [1].

The development of neurosciences shed some light on these topics; it is now widely accepted and more deeply understood that the mind is underpinned by a biological structure that cannot be ignored [2,3]. Considering that both attachment and physical maturation contribute to the development of the individual—and both play an important role in healthy development—we briefly summarize some ideas that underpin the forthcoming reflection.

### 2.2. Attachment

There is considerable evidence of the influence of attachment on child development, including the styles of attachment and personality development [4], the negative outcomes of deprivation [5], healthy and psychopathological development [6], and mentalization and development of self [7].

Humans have a long childhood period, and babies are born incredibly dependent on their caregivers (contrary to other species, including several mammals). The British pediatrician and psychoanalyst Donald Winnicott—who dedicated most of his clinical activity to children—once stated that “there is no such thing as the baby”, along with “a baby alone doesn’t exist. What exists is always a nursing couple” [8]. This iconic sentence of the psychoanalytic field relates to the total dependence of the baby, and alerts us to the need to think about the baby within this relation/attachment (mother–baby dyad). We are born with a considerable potential, but we take much time to become autonomous. The developmental endeavor of becoming autonomous (and the degree of its success) greatly depends on the type (and quality) of the attachment created between the baby and its caregivers.

A mother and child-centered framework is established through the process of the development of the self. The emotional, neurological, and structural puzzle that provides the infant for his future connections is constructed from this plan. Therefore, each person’s attachment style will be determined by the responsiveness, contingency, and persistence of their mother’s replies to their bond requests.

Humans are born with a bonding mechanism that allows them to connect with a caregiver and, from this relationship, develop a connection with the rest of the world. Therefore, the body is the locus of secure bonding expression, the safe place. When a baby feels cherished, he perceives his body as a secure haven. Secure attachment bonds offer emotional support, safety, and availability throughout a human’s developmental history, particularly during trying times and significant moments of change. Infants who are securely attached can anticipate their caregivers’ availability, comprehension, and responsiveness thanks to attachment events. They will consequently feel secure and confident.

According to attachment theory, the stable nature of the attachment bond serves as a vital “emotional buffer” and is essential for completing developmental tasks in adolescence, such as adjusting to bodily changes, developing one’s own identity, or making goals for the future. The baby or toddler can begin to act in ways that involve exploring his environment by feeling secure and confident thanks to the safe bonding response. Initially, the main caregiver figures, followed by additional characters, the world space, and so forth.

The body serves as both the place and the agent of discovery and exploration. The physical body, with its limitations and potential, is a place and agent of pleasure and pain. Bowlby’s initial choice of protection as a “principal” biological function is no longer necessary, as noted in Bowlby’s final collection of lectures from 1988. Instead, the multiple advantages of attachment—such as feeding, learning about the environment, self-regulation, and social interaction—all contribute to its conveying an evolutionary advantage. According to this theory, attachment is not seen as a symptom of immaturity that needs to be overcome, but rather as a normal and healthy trait of individuals throughout the lifespan.

#### The Attachment Behavioral System

An “attachment behavioral system” is hypothesized to contain attachment behaviors. To characterize a species-specific system of behaviors that result in certain predictable consequences, at least one of which contributes to survival and reproductive fitness, Bowlby [9] borrowed the behavioral system notion from ethology. Inherent motivation is a part of the behavioral systems idea.

Children are believed to form attachments to others regardless of whether their physiological demands are satisfied, so there is no need to see attachment as the result of any more fundamental processes or “drive”. Evidence demonstrating that attachment is not caused by links with food, as suggested by secondary drive theories (e.g., [10]), supports this theory, as was already indicated [11,12]. The infant chooses the actions that are most appropriate for that situation and with that caregiver. As they grow, children have access to a wider range of approaches to proximity and learn which ones work best under what conditions. In fact, as Sroufe and Waters [13] noted, this organizational perspective aids in explaining stability in the face of both developmental and environmental changes.

Murray [14] established the concept of “affiliation”, according to Bowlby [9], p. 229: “Under this heading are classified all indications of friendliness and kindness, of the desire to accomplish things in partnership with others”. As a result, it encompasses a far wider range of behaviors than attachment and is not meant to include conduct that is focused on one or a small number of specific figures, which is the characteristic of attachment behavior. Thus, the organization of the biologically driven, survival-promoting desire to be sociable with others is referred to as the *social system*. The likelihood that people will spend at least some of their time with others is a significant expected result of activating this system.

Whenever it comes to what initiates behavior, what stops behavior, and how behaviors are organized, studies of both humans and other primates clearly demonstrate distinctions between the attachment and sociability systems [15,16].

When a child is secure, they want to play and engage socially, so, when a youngster is happy and certain of his or her attachment-whereabouts, they search for a playmate. Additionally, once they find the playmate, they want to interact with them in a playful manner. So, as a conclusion, a secure attachment allows the toddler to feel secure and to explore and places the body as central. Through the lifespan, it will allow engagement in play and sports. The usage of the body replays the early playful moments with primary caregivers.

### 2.3. Motor Behavior

The development of motor behavior heavily depends on maturation. The exploration of the surrounding environment by children requires movement, which is a key element in child development. Babies develop within their mother´s uterus for as long as possible; however, when they are born, despite their innate skills, they are still very “incomplete” and totally dependent on external support. From the rudimentary movements and basic skills of the baby, to the possibility to execute more complex and intentional movements (with higher capacity to act on the surrounding environment), children progressively develop (with more or less harmony) their motor skills, cognition, and language. During the early stages, children considerably increase their motor abilities; in a short period of approximately two years, they learn how to reach and grasp, to sit, to stand and walk, and to chew and talk [17]. Here, the action/movement of the body should be understood as a way of expression (speech), in which an increased physical coordination and motor competency/domain reflect a more unified self.

Through movement, children progress from the total dependency of early childhood to a more active functioning of exploration of the environment. To crawl and, later, to walk are examples of remarkable conquests made by toddlers and allow the first movements away from the secure base (mother). These movements symbolically relate to the earlier essays of autonomy [18]. The motor development allows new opportunities for children to learn about the surrounding world, and the flourishing motor skills instigate developmental changes in perceptual, cognitive, and social domains [19].

### 2.4. Play

As described, during the early stages of their lives, children progressively increase their motor repertoire because of both maturational and interactional processes with their environment. The acquisition of these motor skills, along with cognitive development and construction of psychic apparatus, occurs through play activity [20], which is internally motivated and has a symbolic feature that promotes wellbeing and pleasure.

There are several types of play; these develop and evolve during childhood. The different types of play are not mutually exclusive; actually, they frequently overlap, but one of them tends to be more active/present during a certain period of child development, based on his/her degree of psychic structure. So, different types of play are related to different types of internal functioning. This is the reason why children with different ages tend to choose different types of play which are more in line with their stage of development. During childhood, new types of play overcome the preceding ones, which requires new skills. There are several taxonomies of play; these vary based on different scientific approaches and disciplines. This is a possible definition from developmental psychology:

Play is often defined as activity done for its own sake, characterized by means rather than ends (the process is more important than any end point or goal), flexibility (objects are put in new combinations or roles are acted out in new ways), and positive affect (children often smile, laugh, and say they enjoy it). These criteria contrast play with exploration (focused investigation as a child gets more familiar with a new toy or environment, that may then lead into play), work (which has a definite goal), and games (more organized activities in which there is some goal, typically winning the game). Developmentally, games with rules tend to be common after about 6 years of age, whereas play is very frequent for 2- to 6-year-olds [21].

According to Pellegrini [22], play has four domains: social, locomotor, object-directed, and pretend. Games come later and are more demanding from a social point of view.

Play is very important during child and self-development; there is solid scientific evidence corroborating this [23].

#### 2.4.1. Social Play

Social play is part of the interaction between children and adults (typically parents), and between children and other children. Peek-a-boo play is one of earliest forms of play, in which babies and their parents engage in interactions characterized by some of the hallmarks of play, such as unpredictability, vocalizations, expectations, and positive affect. The quality of the social interaction between children and their parents strongly influences the competence of the children in future interactions with peers and others, due to the social learning and nature of the attachment between parents and children [22].

From a psychoanalytical standpoint, peek-a-boo is based on presence/absence, and relates to the need to be sought and found, as well as the role of being seen in the development of the self. By playing this, children progressively try to elaborate and symbolize the absence, which is a fundamental acquisition for their future social and academic endeavors [24]. Toddlers are thrilled when their parents hide behind a sweatshirt and ask aloud where the baby is, and then they put down the sweatshirt and show their happy face for “having found” him/her. Fragments of this type of play remain during childhood, although it may appear in different manners such as turning lights on and off, or later in the hide and seek game.

#### 2.4.2. Locomotor Play

Locomotor play comes later, and the child’s body is more actively involved; it includes a dimension of physical vigor, and consists in exaggerated and non-functional behaviors and behavioral sequences. According to Pellegrini [22], there are three sub-types of locomotor play, with different age peaks: (1) rhythmic stereotypies (infancy), (2) exercise play (preschool years), and (3) rough-and-tumble play (middle childhood). Locomotor play has physical, cognitive, and social benefits, and there are some gender differences, with boys being more prone to engage in this type of play, especially exercise play and rough-and-tumble play. Although rough-and-tumble play is a sub-type of locomotor play, due to its importance in future sport activities, we describe it with more detail. Rough-and-tumble play is frequent among juvenile mammals (mostly among males), and has an important impact on physical and psychological development. It consists in play fighting (very different from real fighting). Children wrestle, climb over each other, and roll around, among other similar behaviors. Besides developing strength and physical skills, it also allows participants to assess other´s skills and physical strength [22,25]. Rough-and-tumble play fosters both psychological development and social skills; more specifically, it helps children to learn the limits of their strength, to find out what other children will and will not let them do, to develop social relationships due to the change in roles and sort out personal boundaries, and, also importantly, burn off energy and decrease tension.

#### 2.4.3. Object-Directed Play

In object-directed play, children use objects (extensions of their bodies) when playing; they discover novel uses for objects, and these can be used in exploration, in play (with objects), in construction, and in tool use [26]. The use of objects is connected with social network and group structure.

#### 2.4.4. Pretend Play

Pretend play—also called imaginative play or dramatic play—happens when children use their imagination to enact scenarios (experienced, seen, and/or fantasized). It may progress from solitary to social pretending with others, and according to Pellegrini [22] there are four components of pretending: (1) decontextualized behavior, (2) self–other relations, (3) sequential combinations, and (4) object substitution.

Pretend play is rooted in early interactions with adults, especially the child’s mother (attachment), and may be exclusive to humans. It becomes more intense with the rise of the Oedipal complex. This type of play allows the possibility of movements towards identifications, the deeper acknowledgement of sexual differences between boys and girls, and other important aspects of construction of the identity [24]. Here, children enact different roles (e.g., super-hero, princess) within an interpersonal scenario with peers and other important figures. This type of pretend play allows children to dramatize his/her inner world, speaking about him/herself through others. Some gender differences should be noticed; boys and girls tend to choose different characters, which becomes more pronounced at this stage.

The definitions of play and games differ; the latter develop later in childhood, are governed by specific rules based on deduction [27], and the aim is usually to win. During childhood, children tend to have an idealized idea of their parents, and perceive them as omnipotent entities who know everything, and are able to solve any problem. This (normal) idealization is rooted in the need to feel safe in the face of anxieties (and vulnerability). In healthy development, the idealization (and illusion) progressively decreases, allowing children to abandon their omnipotence (self-centered interpretation of the world) and to incorporate social norms and rules, provided by social order; as a result, children can then access more abstract thinking. In this phase, the game emerges; it reflects a higher differentiation between psychic structures (id, ego, superego), and is underpinned by both competition and measurement of skills between the participants (peers function as a mirror).

Play (and later, games) have an important role in child development. Play allows—among other things—the expression of internal conflicts (some of them normal, others more prone to become psychopathology), and children change from passivity to activity, allowing them to act upon their surrounding reality. The later engagement in games is more demanding; due to their standardized characteristics, to properly engage in games, children must submit to social rules and norms. We consider that the earlier forms of play and games—along with specialization of basic motor skills—underpin sport activities. More on this topic will be discussed later.

#### 2.4.5. Importance of Play in Natural Environments

Despite its benefits for child development, the amount of time spent by children on outdoor free play (unstructured activities) has considerably decreased in current societies [28,29], contributing to more sedentary lifestyles, disconnected from the natural world [30].

Unstructured outdoor play has several benefits in child development, health, and wellbeing, including physical development, self-regulation and attention, communication and social development, cognitive development and creativity [29], and imagination and reasoning abilities [31]. Outdoor play also contributes to fostering physical exercise (and energy bust), to improve social and emotional development [18] and may contribute to establishing rapport between parents and their children [31]. Due to its benefits, outdoor play has been considered in educational approaches [28,32].

## 3. PART II. Actual Constraints on Child Development

### 3.1. Settings

Nowadays, mostly in Western societies, child movement is frequently (and early) not accepted and contained. The COVID-19 pandemic has contributed to the decrease in child movement due to lockdown and other preventive measures that resulted in a decrease in outdoor areas to play, and increased the physical distance between people [33]. In their review article, Kourti et. al. [34] suggest that play habits changed during the COVID-19 pandemic (and teachers were concerned about it); the authors analyzed 17 articles from Europe and North America, and concluded that outdoor play decreased during the pandemic, while indoor play and videogames/screen use increased. However, even before the pandemic, some contexts were identified in which child movement was already being felt as unacceptable. Here, we highlight three of them.

#### 3.1.1. Home

At home, within the family environment, when parenting styles lead to insecure attachment styles, the natural movement of infants—which is associated with (normal) aggressiveness, dirtiness, excitement, and noise—is perceived as uncomfortable to some parents [24]. The normal movement of children is poorly understood and contained by an external environment that aims to impose excessive order and tidiness, going in the opposite direction of children’s needs. When this happens, as submission is gradually imposed, children (especially male) are pushed to become “good”: quiet, clean and “well-behaved”. Here, when children behave accordingly, their submission tends to be reinforced by parents (and, as a result, the opposite behavior is rejected). For girls, it is expected, even in an unconscious way, that they behave, stay quiet, and play with dolls or house tools.

The increased (and often excessive) use of electronic devices among children also contributes to the decrease in play. These devices keep children quiet in front of a screen; it becomes very useful when parents wish for their children to demand less attention from them. The high number of children who cannot be seated at the dinner table with their parents (and perhaps siblings) without having an electronic device to play is impressive. Despite the positive aspects of technology, the excessive use of electronic devices by children is a red flag regarding their healthy development. The stimulus of electronic devices is excessive. This aspect leads to a growing necessity of fast pleasure and immediate reward. In an older child, the world is “expected” to be pleasurable and satisfying. So, frustration and need for continuous reward are a predicable factor of anxiety and aggressiveness among older children.

According to Desmurget [35], the excessive use of electronic devices by children may result in negative outcomes regarding their physical health (obesity, heart diseases, lower average life expectancy), behavior (aggressiveness, depression, anxiety), and intellectual skills (language, concentration, and memory). Some major international organizations have also focused their attention on the impact of excessive use of electronic devices by youngsters [36,37].

#### 3.1.2. School

The first major separation of children from their nuclear families and homes occurs when they go to school. Even those children who have been in kindergarten before going to school now have to face a more structured and demanding environment. Children face challenges such as the need to develop their skills, to learn school material (within a specific time), to make friends, and to progressively become less dependent of adults—to grow. Obviously, attachment issues have a strong influence on their success in these demanding tasks.

School is a very important context and allows children to grow and develop their skills. However, their natural movement is also often poorly understood and tolerated. Children are often asked to keep quiet, and to listen to a teacher for long periods of time, in an environment often marked by an obsessive analysis of school performance (grades) and accumulation of material rather than fostering understanding [18,38]. Education has become more technical and “pseudo-cumulative”, with emphasis being placed on “skill acquisition” rather than fundamental knowledge and fostering the development of the capacity of children to think about themselves and how the world works.

Today´s school asks children to carefully listen to what their teacher tells them; however, children are being raised in a world where the value of words has decreased considerably over time, and children are constantly a target of visual, immediate, and short-term stimuli (electronic devices, social media, and others). As a result, the opportunities to appreciate silence, to contemplate, to feel, to listen and be listened to, and to have relationships with an adult caregiver that foster thinking about how the world works and thinking about themselves (which is time consuming), have decreased [24].

#### 3.1.3. Outdoor (Street)

The lack of play in cities and streets is also a constraint. During recent decades, outdoor (street) free play has been gradually replaced by standardized and artificial activities [39], in which children are oriented by adults in predefined and goal-oriented activities. Free time has become a paradox; children are often asked to do something in time periods that are supposed to be free (and include the possibility not to do anything). Laziness among children is often necessary and important; when children feel bored, they tend to use creative solutions to amuse themselves. This often allows games and play to arise. In urban areas (where most families live), the presence of children playing on the street has significantly decreased. They mostly live locked at home and are transported by adults from point A to point B all the time. Today, the street, perceived as a place to play with other children, practically does not exist.

Parents, schools, and urban areas often have difficulties to contain and to provide meaning to child movement. The natural benefits of play are being replaced by artificial programs (often seen at school) to “teach empathy”, to “develop social skills”, and/or to “learn about emotions”.

### 3.2. Psychopathology

The verbal skills and the capacity of children to access symbolic thought, when compared to adults, are, obviously, lower (and still under development). When play decreases among children, psychopathology tends to arise [39,40]. Clinicians who work with children often observe infants’ difficulties being expressed through their bodies. We do not intend to cover child psychopathology in detail; for the purpose of this article, we divided the problem into two groups: children whose (normal) aggressiveness becomes inhibited (lack of movement) and, at the opposite pole, those whose who are characterized by excess of movement (due to the lack of containment).

#### 3.2.1. Lack of Movement: Inhibition of Aggressiveness

Inhibited children tend to be more submissive, formal, and “hyper-mature” (behaviors often reinforced by their parents and teachers), kinds of mini-adults who lack spontaneity and have difficulties with imagination and fantasy. These children are usually perceived as well-behaved and a role model at school but have difficulties in making friends and in free spontaneous play.

Inhibited children are much less often signaled at school (where many cases of primary diagnosis are spotted) because they do not disturb adults. The preoccupations around these children are often connected with their difficulties with peers (e.g., bullying victimization), a sudden (and not expected) break in their academic performance, or abrupt change in behavior. The expression of normal aggressiveness often fails in submissive children, not allowing them to be properly assertive. The parents of inhibited children (especially boys) often seek sports activity for their children, expecting them to learn there how to defend themselves (e.g., fighting sports), or to make friends to compensate for the loneliness felt at school due to peer rejection.

#### 3.2.2. Excess of Movement: ADHD and Related Issues

The opposite pole to inhibited children are those children who cannot calm down and are constantly excited and hyperkinetic. In these cases, the movement is disorganized and expresses internal conflict and/or nameless threats. The excess of excitement (that should not be confused with normal motor activity) often reflects a poorly harmonious evolution in child development. Inhibited children often have problems with expressing normal aggressiveness; on the other hand, hyperkinetic children (mostly boys) often have problems related to lack of containment. These children are often labeled as suffering from attention deficit hyperactivity disorder (ADHD).

Interestingly, these children tend to behave differently if they are alone, with parents, in school, or with someone who they do not know. Their behavior also differs considerably when they are with peers (more agitated) or in a one-on-one situation with an adult (they tend to regulate more easily), especially a male figure. Finally, very stimulating environments and situations in which children are freer to control their learning rhythms, and situations in which they are being paid to do tasks, also contribute to the ability of ADHD children to regulate themselves [24]. These variables allow us to think about the role of internal working models in ADHD, as well as the importance sports may have in the regulation of these children; those (especially boys) who have difficulties in impulse control and affect regulation often find sports a positive environment, and may strongly benefit from such activities.

The benefits of sports for children at both poles will be described in more detail later in this essay.

## 4. PART III. Role of Sports in Child Development

### 4.1. How Sports Foster Healthy Development

Sport participation fosters overall wellbeing of children [41]. Efforts have been made to develop models to describe how sports foster human development. Here, we briefly describe three proposals, based on different approaches, and considering different variables.

#### 4.1.1. Developmental Model of Sport Participation

The Developmental Model of Sport Participation (DMSP) is based on theoretical and empirical data, and aims to describe the processes, pathways, and outcomes related to the participation of children and adolescents in sports [42]. The DMSP considers the importance of appropriate training considering the age of the participants and their physical and psycho-social development. It identifies three trajectories towards grassroots and elite performance: (1) sampling years (age 6–12), (2) specializing years (age 13–15), and (3) investment years (age 16+). The DMSP considers early diversification (taking part in different sports) to foster the development of general motor skills. It also calls attention to the benefits of a high amount of deliberate play, and a low amount of deliberate practice, during sampling years.

#### 4.1.2. Personal Assets Framework

The Personal Assets Framework [43] aims to describe the mechanisms of positive youth development in sports. This model considers that sport experiences should be analyzed based on three dynamic elements: (1) type of activities (What?), (2) quality of relationships (Who?), and (3) setting (Where?).

According to Côté, Murata, and Martin [41], the type of activity emphasizes the need to tackle early specialization due to its negative impacts on children (see [44]), and to foster play during childhood while progressively increasing practice as youngsters grow older and approach adolescence. The quality of the relationships includes interpersonal relationships, team dynamics, and the broader social environment; different types of relations (attachments), such as adult–child, child–child (peers), child–club, individual, and group relations, should be considered. Finally, the settings include structures that provide physical support for the sport activities.

#### 4.1.3. Long-Term Athlete Development

Long-term athlete development is a developmental model underpinning a considerable amount of research, that aims to foster participation in sports and physical activity throughout life, by describing what people should be doing at certain ages, in sports. The idea of doing the right things at the right time, in sports participation, is described in nine stages, from childhood to adulthood, divided by gender. The stages consider the physical, intellectual, cognitive, and moral development of the individuals within each category, and are divided into: (1) Active Start (age 0–6), FUNdamentals (age 6–9), Learning to Train (age 9–12), Training to Train (age 12–16), Learning to Compete (age 16–18), Training to Compete (age 18–21), Learning to Win (age 20–23), Winning for a Living (age 24+), and finally Active for Life (any age). The first three stages are mostly focused on physical literacy and development of basic motor skills through play; these underpin further endeavor in sports, from being active in life to competition [45].

Despite their differences, all three models overlap regarding the need to foster free play during the early stages of athlete development, and describe its benefits for motor, social, and psychological development. Children (especially boys) easily engage in sport activities, and these can play an important role in child development.

### 4.2. Maltreatment and Abuse in Sports

Sports are generally perceived as positive, and their potential benefits for children are well known; as a result, parents, teachers, and physicians, among others, often foster the engagement of children in sport activities. However, participation in sports may also result in negative outcomes; despite the potential to foster human development, several types of abuse may occur during sport activities. Abuse in sports settings has been studied—especially in the last couple of decades—including research on bullying [46], coach emotional abuse [47], early specialization [44], sexual abuse and harassment [48], and different types of interpersonal violence [49], among others. The results of the studies, as well as the public knowledge of some major scandals, called attention to the need to protect young athletes from abuse. Safeguarding in sports has progressively become a growing concern.

When parents leave their kids in sport clubs, they expect them to be in a safe environment, and engaged in a positive activity with peers, regulated by a responsible adult. However, as stated before, different types of abuse may occur. Some children may be (re)victimized in sports, rather than finding a healthy environment that helps them to grow and overcome their difficulties. We can then ask, are sports positive for children?

### 4.3. Does the Engagement in Sports Foster Child Development?

The right answer to this question is probably that sports are a very powerful tool to foster human development, and children may find many positive opportunities to develop physical and emotional skills, and to improve themselves constantly. However—and despite their potential—the quality of the engagement in sports greatly relies on the quality of the relationships (attachments) established with significant others within this setting (and with sport itself). We can conclude that merely practicing sports cannot ensure positive outcomes, but if sports activities are properly conducted and adapted to children’s developmental stages, then sports can be an outstanding tool to foster positive development of children and youth.

Being a competent child´s sport coach is high skilled and demanding activity. These coaches must have knowledge about sports training, but also about child development. To become a role model, these coaches must understand the children´s needs, and always focus their attention on their best interests.

### 4.4. Understanding Sports

Those who are/were athletes, and even those who do not practice sports and prefer to merely watch, are often excited by the accomplishments of athletes, or become profoundly sad and disappointed when their admired athlete or team loses a match/game. People often celebrate a point/goal/win in enthusiastic ways, rarely seen in other areas of their lives. Sports promote fights and disputes, but also companionship between people. The passion is often used as an explanation to explain such apparently weird behaviors. Besides that, sport activities are universal, which calls attention to their importance. Actually, sports are an ancestral activity, deeply rooted in all cultures, underpinned by play, and include corporeality and intra- and interpersonal relationships within a competition setting. Sports have been studied by several scientific areas, with different aims and methods. Historical analyses of paintings on the walls of caves, artifacts, and manuscripts, among other records and sources of information, describe how sport activities have always been spread all over the world, and have always had an important social role [50,51,52,53]. There seems to be a common base for most sports; modern sports are underpinned by the ancient ones and—despite some differences due to geographical, cultural, and chronological variables—there is a considerable overlap between them.

Sport sciences have focused their attention on improving the performance of athletes (including many efforts of sport psychology). However, other approaches focused mostly on understanding sports, by analyzing their social and symbolic aspects; among these are included sociology, anthropology, and social psychology. Regarding psychoanalysis—perhaps surprisingly—there is a lack of research on sports [54,55]. The involvement and participation in sports are not purely rational; unconscious drives and life and death instincts play an important role that helps to understand why humans become involved in sports, and the reason for the great amount of pleasure from such activity [56]. To better understand sports and their “passion”, we must look at the symbolic meanings of sports activity. The psychoanalysis approach to sports considers topics such as aggressiveness, narcissism [54,57], sexuality, and attachment themes throughout the human life cycle [57], among others.

#### Basic Elements of Sports

Sports are not easy to define and include under the same concept. Several operational definitions and taxonomies have been proposed, and none of them is totally satisfying. We do not intend to develop such a conceptual discussion. For the purpose of this article, we will adopt a definition that—despite its limitations—has received a large consensus. The European Sports Charter ([58] https://rm.coe.int/16804c9dbb (accessed on 1 January 2023), in Article 2, defines sport as: *“all forms of physical activity which, through casual or organised participation, aim at expressing or improving physical fitness and mental well-being, forming social relationships or obtaining results in competition at all levels”.*

Despite the differences between different sports (and sports definitions), they are all underpinned by three basic elements: (body) movement, play, and competition (games).

(a) Movement

Sports result from the specialization of basic movements such as walking, running, jumping, kicking, grabbing/throwing, diving, climbing, etc. As a result, the body plays a major role in sports activities. Extension materials should also be considered (these are often used in child play); here, the body receives a “new part”, an extension, perceived as a continuity of the body itself (hockey or tennis are good examples).

The observation and analysis of the movement should not be based exclusively on sports techniques based on a musculoskeletal approach; it should also include the imprints from the psyche. The way children move and perceive their body is strongly related to their psychological development and attachment issues. Sports should therefore be perceived as an important area to provide meaning to movement, through a relation with another (bond). This is particularly important for youth sports training and young children who might suffer from inhibition of aggressiveness, ADHD, and other related issues.

(b) Play

Sports occur in a context of play with rules (games). They are underpinned by competition, and include ludic, symbolic, and pleasurable aspects. The understanding of the symbolic features of sports (and their potential to transform), as well as the working models of children, allows is to use sports in the service of child development. Sports allow the sublimation of aggressiveness in adequate ways, as well as impulses and unconscious desires.

(c) Competition

Competition is inherent to sports, and the engagement in these activities allows children to express normal aggressiveness. By playing sports, passiveness is turned into activeness, and children may symbolically domain others, and simulate their destruction (fantasy). The unconscious elements involved in sports may be expressed in adequate and progressively integrated ways (rather than being projected and acted out).

The binomials inside/outside, activity/passivity, and attack/defense interconnect and mutually influence each other. Many sports symbolically relate to attack and defense, to kill or die. The notions of space, territory, and target (own and opponents) should also be considered when analyzing sports. Besides motor skills, cognitive and spontaneous domains also play an important role and should not be ignored. The cognitive domain relates to planning and discipline; the representation (thinking and imagining) of an action before acting and the discipline to execute an action plan are always present in sports and extend to other areas of children’s lives (such as school). The spontaneous domain relates to freedom, to a creative flow, complementary to the cognitive domain.

### 4.5. Positive Outcomes

Here, we aim to analyze the use of sports as a tool to foster child healthy development. To do so, we divided the positive outcomes of sports into three major categories: (1) assigning meaning to body movement, (2) corporeality, fantasy, and play, and (3) ethics: positive identifications and sense of belonging. These areas are discussed and analyzed in the following.

#### 4.5.1. Assigning Meaning to Body Movement

The brain has seven primary process emotional systems: SEEKING, SADNESS, FEAR, LUST, CARE, ANGER and PLAY; these are strongly linked to psychiatric disorders. The PLAY/JOY system stimulates young animals to engage in physical activities such as those described in rough-and-tumble play (wrestling, running, chasing each other), which helps young mammals to learn social limits and develop impulse control. If children do not play (or play less than needed), the development of their brain may be impaired, resulting in consequences in their maturation [3,59]. According to Panksepp [40], the increasing prevalence of ADHD among children may be related to the decreased opportunities for preschool children to engage more often in natural self-generated social play, because this type of play facilitates behavioral inhibition, while psychostimulants reduce playfulness.

When children are medicated for ADHD, many decrease their motor activity, but tend to feel apathy [24].

Rough-and-tumble play is also very common in childhood, especially among males; it contributes, among other aspects, to the development of social skills and capacity to increase concentration [60]. Both types of play described here underpin sports. Considering the actual decrease in play and generalized restrictions to movement among children, sports clubs became even more important to allow children to play (and move). If training is properly managed, and playfulness overcomes the development of specific motor skills, sports clubs can play a major role in assigning meaning to movement. The developmental models of young athletes overlap in the need to emphasize play and unstructured activities in early phases (see the section How Sports Foster Healthy Development). Unfortunately, many coaches of young athletes overemphasize the need to win in the short term, rather than focusing on the playfulness of sports, and on the development of children’s motor repertoire during their early training phases. Panksepp [40] proposes that play “sanctuaries” for children who suffer from ADHD should be established as an alternative for psychostimulants that reduce play. The author considers that this action would foster frontal lobe maturation and promote the healthy development of pro-social minds.

However, it is not only children who are hyperkinetic who benefit from sports. Those children who have difficulties in expressing normal aggressiveness and become too submissive may also find in sports an important setting to tackle their difficulties. Aggressiveness is natural and important; children should learn how to use and integrate it in proper ways (e.g., assertiveness). When aggressiveness is not integrated, and is turned towards the self, difficulties can arise. These children may become too mechanical, functional, and (pathologically) normative [61]. Children who suffer from obesity, dyspraxia, impairments in lateralization, and tics may also benefit from their engagement in sport activities, because they enhance body scheme and spatial–temporal representation, improve physical skills, and allow them to express aggressiveness in adequate ways.

Many difficulties of children are expressed through the body, and modern lifestyles contribute to the decrease in free play and constraint of movement. Sports can play an important role in stimulating play and movement of young children, fostering their motor, social, and psychological development. To do so, training practices should consider age appropriate activities, including free play and unstructured activities in early stages. The specialization of motor skills should be gradual, with training focused on enhancing performance, and coaches should be (more) aware of the needs of the children. In those cases of children who are inhibited or hyperkinetic, it is important understand their movement as a reflection of their internal world (and conflicts).

#### 4.5.2. Corporeality, Fantasy, and Play

Advances in technology have brought major developments and improved the quality of lives worldwide. However, the virtual experience has also supplanted a large amount of embodied experience, which makes involvement in sports (playing) an important way to stay connected with a deep and ancestral part of human life [62].

Sports have a connection with psychoanalysis due to both considering interrelationships among intrapsychic, interpersonal, and social realms [57]. Involvement in sport activities includes the use of body expression (motor skills), within a ritual that considers both internal and external rules to regulate it (Freudian id, ego, and superego). According to Free [56], the overall system of allowances and prohibitions in sport activities may be perceived as an analogy to the Oedipal complex; most sports allow the symbolic physical expression of unconscious desires for maternal possession (which can be seen in the efforts to conquer territory, to score goals, to penetrate lines), but also require internalized discipline associated with parental authority (super ego). The pre-Oedipal functioning relates to indulgence, libido, and aggressive instincts, while the post-Oedipal regards aspects such as maturity, discipline, submission to social reality, and norms.

The internal working models of children influence how they perceive sports, and how they engage in such activities. The questions about “what” and “how” children play should be considered.

What a child plays is related to the sport chosen by him/her. What does the child try to symbolically express by playing that sport? *How* a child plays is related to the type of attachments he/she establishes with sports, peers, coaches, and competition. Some children do not allow themselves to win (due to the unconscious fantasy of destroying the other by doing so), while others cannot stand to lose (felt like a dangerous attack to their self-esteem). Learning how to “play the game” is very important for youngsters who engage in sports. By playing the game, we refer to learning how to relate to others in a positive way. More on this is developed in the next section.

#### 4.5.3. Ethics: Positive Identifications and Sense of Belonging

According to Bonovitz [63], sports play an important role in the need for adolescents to search for idealization and idealized objects, for example, sport stars, allowing them to identify with these heroes, and foster their efforts for further separation from their parental figures. This positive idealization and identification reinforce the sense of identity of youngsters, within a social environment outside their families, and not directly linked with them.

By identifying with their sport heroes, youngsters may seek self-improvement and personal transcendence. In a broader sense, this is an ethical perspective of the utility of sports, that may work as a significant means to foster the identification with a competent, strong, fair, and resilient person, who works hard on a daily basis to overcome obstacles and become a successful hero. This perspective overlaps with the idea of the Jungian archetype of the hero [64].

Sports activity relies mostly on self-improvement and self-monitoring. There is the underpinning idea of competition: the best win and become champions—*Citius, Altius, Fortius*—*Communis* (Olympic Motto—Communis em 2021 [65]). The pursuit of things of value within a social setting (that include cooperation, rivalry, and competition) is important, and relates to the idea of a hierarchy. Hierarchies between people foster people to improve themselves so they can reach their goals and achievements. The idea of the champion is based on someone who is able to achieve something special; to do so, those who want to become champions must adopt the carpe diem ideal of using every day to improve him/herself (following a plan, with predefined goals). Although winning a game/competition is important, and athletes should focus their attention on it, and give their best to be successful, that game/competition is “just” one part of a much wider endeavor, that inevitably includes victories and losses, glory and frustration. This is a metaphor for life, and that is why winning each game is important but, at the same time, it is not. In the end, what might be important to teach children through sports is that the whole process, the whole sport life, is the most important, and to enhance the chances of being successful, one must learn how the world works, and how to play well with others.

Hierarchies result from the need to create a social organization, based on perceived value of their elements, that organize social relations. Hierarchies and power are frequently (and wrongly) associated with coercion and abuse. It is the misuse of power (abuse) that makes hierarchies non-functional: when elements arise from another factor rather than competence, then the hierarchy becomes tyrannical. The ideal of sports is to promote the best, based on their skills and competence, regardless their background, country, or any other factor that is not competence. Poor coaching perceives hierarchies wrongly, and tends to organize youth sport activities around two opposite poles: (1) flat hierarchies, and (2) win-at-all-costs mentality. Both approaches have flaws.

Flat hierarchies, considering that everyone wins despite their performance, are usually justified as a way to avoid negative feelings of frustration among youngsters, resulting from not being the best/champion. This approach is frequently fostered by those who consider competition a negative thing. We will try to explain why this approach is wrong.

First of all, if there is no competition, then we are not talking about sports. Competition is part of sports (and life in general). Avoiding competition is not a good strategy (especially in a long-term perspective) because it does not push children forward in order to pursue their goals. When this approach is undertaken, children see themselves as the Dodo Bird in Alice in Wonderland: despite their performance, everyone wins, and everyone has presents. When adults tell children that they win, no matter what they do (and their result), and that everything they do is always fantastic, children perceive their accomplishments as fake, and the judgement of adults as not reliable. This continued approach may contribute to the poor development of narcissism of narcissism, and the consequent feeling of not having real value. In these situations, children tend to remain in a pre-Oedipal state, not facing the social rules and personal limitations, and not being encouraged to expand their skills. The idea of keeping children in a non-confrontational ideal world does not allow them to grow up and accept their own (and others’) flaws and limitations. Here, it is important to have the role of the masculine, that helps to separate children from their protective mothers, and foster their endeavor to conquer unexplored territories.

Avoiding competition is a poor approach; however, the opposite pole of *winning at all costs* may not be the best option either. When adults teach children that winning at all costs is the thing to pursue, they are missing the point of the utility of sports. It is important to explain to children that the most important thing is to learn how to play the game. This does not mean that winning the game is not important, but it is more important to learn how to develop cooperation and to compete in the long term. The most important thing is to pursue this long-term goal (and sometimes to sacrifice short-term goals of winning the game) because the present competition is just a piece (and the beginning) of something bigger, longer, and more important.

Children should be trained not to necessarily win isolated games, but to improve their performance and become better, game after game (like in life). This idea is underpinned by the Olympic Motto and relates to the development of character or an attitude towards life, based on pursuing valuable things and playing well with others (in sports and life), which will increase the possibility of having success in life. To do so, sometimes one needs to sacrifice present satisfaction (winning now), and to progressively learn how to delay pleasure and build solid endeavors. By success, here, we refer to the improvement of relationships with others, by learning to metaphorically “play the game”, which will result in more positive relationships with peers, adults, and the whole community. To be successful is to learn to develop strategies to go well in life in the long term; to do so, children must identify themselves with role models not solely based on their skills, but mostly on their attitude towards sports and others.

Besides fostering active engagement over passivity, intense involvement in sports also provides a feeling of community and sense of identification [66]. According to Free [56], formally organized games and sports are reality-adapted play (post-Oedipal) that foster mutual identification between athletes, and observing rules is more significant than the victory itself. Peer relations become more and more important as children grow, peaking in adolescence, and engagement in sports can also play an important role here.

## 5. Conclusions

Sport activities include topics such as motor behavior (movement), unconscious motivation, conflict, and fantasy. In sports, youngsters find a way to express themselves; they allow them to confront their limitations while, at the same time, they have a huge potential for transcendence and improving resilience.

Considering the decrease in free and outdoor play in Western societies, sports become more and more important to provide meaning to child movement expressed through their bodies. Sports can be a powerful tool to foster human development. To do so, it is important to understand the relational dynamics of children, the quality of their attachments, and to know how to interpret the symbolic meaning of their behaviors. This humanistic approach to sports is the opposite of a considerable part of modern youth sports training, focused exclusively (and obsessively) on sports performance and competition. The obsession with sports performance among youngsters (as often happens at school) contributes for poor sport practices, and makes sports less fun for the athletes.

Despite the enormous potential of sports to foster child development, the quality of sport experience depends largely on the type of attachments and relationships established between the youngsters and other stakeholders in their sports environment: coach, peers, parents, club, and community. The active and continued engagement in sports fosters motor, social, and (healthier) psychological development, and should therefore be considered in youth education.

## Data Availability

Data hasn’t been collected.

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
