# Peer review of "Movement, Play, and Games—An Essay about Youth Sports and Its Benefits for Human Development"

_healthcare, 2023, doi:10.3390/healthcare11040493_

Round 1

Reviewer 1 Report

The submitted work intends to develop an essay about the importance of play and youth sports on child development. Although a relevant topic, I have found some issues that I will express during my commentaries. Hopefully, they will serve the authors in the improvement of the developed work.

General notes

I have found the work well written and articulating several topics usually disconnected or poorly explored when thinking about the role of sports in children’s development. I commend the authors on the topic exploration performed.

An issue detected during my reading pertains to the structure of the essay. Sometimes it is not clear why some sections or headings are presented or their order. See for example the title in Part I: “CHILD DEVELOPMENT. PLAY AND MOTOR BEHAVIOUR”. Although the structure was explained in the introduction, the section body and attachment seem to appear rather disconnected given the title. I believe that i) the Part I title needs refinement, or ii) the body and attachment sections should appear as an “introduction” to the core themes of Part I. Additionally, in this heading title, as in the introduction section, motor behavior appears before play, an order different from the one in this section. Altogether, some refinements are in need to better transition the explored themes.

Additionally, sometimes there seem to exist two approaches to presenting the information. More often, the authors develop an extensive portion of the text presenting information and scientific claims without any references; in other moments, the writing seems more related to a usual scientific and “traditional” paper, where claims and relevant information are supported by citations. I believe that i) some effort in the uniformization of how the information and respective citations are used could be developed, and ii) more scientific support for some claims is needed. I understand that the nature of an essay heavily depends on the author’s considerations and thoughts on relevant themes and thus is not accountable for referencing. Thus, please consider this commentary more as a suggestion than a critical issue.

Examples:

Lines 65 to 81: Several topic explorations, conceptualizations, and relations are not properly referenced. The three used references do not allow to understand which reference supports those claims. Additionally, updated information on the topic exists, which would reinforce the author’s arguments.  

Another similar example; lines 414 to 431.

Opposing examples: lines 86 to 90; 591 to 602; 640 to 655

Abstract

The sentence starting with "Nowadays" is very long and hard to follow. Some reprashing would help to pass the essence of the essay, particularly relevant given that the abstract is often the first section to be explored by the readers.

"... and the ways to express suffer". What does this mean?

Keywords: Why “mental health”? Why not youth health? If you would like to follow general rules for keyword selection, please do not use terms that are already in the title; search whenever possible for MESH terms.

Introduction

Line 42: This sentence says the same as the one before. It seems redundant.

Part I

Line 90: Typo on the citation

Line 99: “This all process”? Typo?

The section between lines 91 and 123, as for section 1.2.1 (e.g., lines 164 to 171), seems rather long to express an idea (the role of attachment for environment exploration and social system as a booster for movement-based interactions) that will posteriorly be associated with motor behavior.

Please check the sentence in lines 172-174. Something is wrong with the phrasing.

Line 179 “if such a analysis is valid”. Typo

Line 180 “Throw”? Through? Additionally, the sentence needs revision for commas.

Line 238: “The peek-a-boo play is one of earliest forms of play, in which babies and their mothers engage…”. I understand that this sentence comes in accordance with the mother-baby dyad mentioned in the attachment section, however, here, there seems to be a bias by the authors, claiming that only the mother performs this activity. Curiously, in the sentence before, it is recognized that both parents develop interaction with their children. Perhaps clarifying that fathers (or more precisely, the other parent besides the mother; search for inclusive language) also may play peek-a-boo would resolve the issue.

Part II

Line 317-319: “The COVID-19 pandemic has contributed to the decrease of child movement due to lockdown and other preventive measures that resulted in a decrease of outdoor areas to play”. Do you have support for this claim? Given that the authors are Portuguese, I suggest reading the Portuguese report REACT-COVID 2.0 (pag. 8), where, in adults, a slight increase in physical activity was detected. Given that physical activity was one of the motives people were allowed to get out of the house during the lockdown, I am not sure that in children, and all countries, this claim is true. Perhaps reducing the assertiveness of the sentence would better reflect some existing disparities.

Line 343-345: This is one of the examples where a citation would be needed. Otherwise, it may just be the opinion of the authors, based on logical assumptions.

Lines 370-372: The sentence needs an ending, as it is, it is not understandable.

Lines 386-388: Another example needing a citation given that this does not result from the reflection made in the previous paragraphs.

Line 390: “though”? Perhaps thought.

Section 2.2.1: Is this an opinion of the authors or a classification represented in the literature? It is not clear.

Part III

Line 455: To maintain coherence, change “where” to Where.

Line 636-638: The message in this sentence is very relevant, but has some issues with the English (redundancy and typo)

Lines 747-752. What does “enable channelling homosexual urges into teams,” mean?

Conclusion

Line 757: Where was “unconscious motivation” explored in the essay?

Lines 762-764: Once again, this was not clearly explored throughout the essay. Why use expressions, terms, concepts, and ideas not explored during the paper (e.g., language, space-temporal representations)?

Author Response

Reviewer 1

The submitted work intends to develop an essay about the importance of play and youth sports on child development. Although a relevant topic, I have found some issues that I will express during my commentaries. Hopefully, they will serve the authors in the improvement of the developed work.

General notes

I have found the work well written and articulating several topics usually disconnected or poorly explored when thinking about the role of sports in children’s development. I commend the authors on the topic exploration performed.

An issue detected during my reading pertains to the structure of the essay. Sometimes it is not clear why some sections or headings are presented or their order. See for example the title in Part I: “CHILD DEVELOPMENT. PLAY AND MOTOR BEHAVIOUR”. Although the structure was explained in the introduction, the section body and attachment seem to appear rather disconnected given the title. I believe that i) the Part I title needs refinement, or ii) the body and attachment sections should appear as an “introduction” to the core themes of Part I. Additionally, in this heading title, as in the introduction section, motor behavior appears before play, an order different from the one in this section. Altogether, some refinements are in need to better transition the explored themes.

 We understand the comment; however, we find hard to change the structure (based on the comments of all reviewers). We believe that the manuscript follows a logic chain of thoughts (although we appreciate and understand your comment).

::::::::::::::::::::::::::::::::::::::::::::::::::::::::::::::::::::::::::::::::::::::::::::::::::::::::::::::::::::::::::::::::::::::::::::::::::::::::::::::::::::::::

Additionally, sometimes there seem to exist two approaches to presenting the information. More often, the authors develop an extensive portion of the text presenting information and scientific claims without any references; in other moments, the writing seems more related to a usual scientific and “traditional” paper, where claims and relevant information are supported by citations. I believe that i) some effort in the uniformization of how the information and respective citations are used could be developed, and ii) more scientific support for some claims is needed. I understand that the nature of an essay heavily depends on the author’s considerations and thoughts on relevant themes and thus is not accountable for referencing. Thus, please consider this commentary more as a suggestion than a critical issue.

We checked each of your suggestions and changed accordingly.

Examples:

Lines 65 to 81: Several topic explorations, conceptualizations, and relations are not properly referenced. The three used references do not allow to understand which reference supports those claims. Additionally, updated information on the topic exists, which would reinforce the author’s arguments. 

 Rephrased. 

Another similar example; lines 414 to 431.

 Rephrased. 

Opposing examples: lines 86 to 90; 591 to 602; 640 to 655

The theoretical rational of the arguments was improved. More references were added in some section (like the above mentioned).

::::::::::::::::::::::::::::::::::::::::::::::::::::::::::::::::::::::::::::::::::::::::::::::::::::::::::::::::::::::::::::::::::::::::::::::::::::::::::::::::::::::::

Abstract

The sentence starting with "Nowadays" is very long and hard to follow. Some reprashing would help to pass the essence of the essay, particularly relevant given that the abstract is often the first section to be explored by the readers.

Rephrased!

"... and the ways to express suffer". What does this mean?

Rephrased!

Keywords: Why “mental health”? Why not youth health? If you would like to follow general rules for keyword selection, please do not use terms that are already in the title; search whenever possible for MESH terms.

“Mental Health” changed to “youth health”

Introduction

Line 42: This sentence says the same as the one before. It seems redundant.

Deleted

Part I

Line 90: Typo on the citation

Improved

Line 99: “This all process”? Typo?

Rephrased to: The developmental endeavor of becoming autonomous

The section between lines 91 and 123, as for section 1.2.1 (e.g., lines 164 to 171), seems rather long to express an idea (the role of attachment for environment exploration and social system as a booster for movement-based interactions) that will posteriorly be associated with motor behavior.

Rephrased

Please check the sentence in lines 172-174. Something is wrong with the phrasing.

Rephrased

Line 179 “if such a analysis is valid”. Typo

Rephrased

Line 180 “Throw”? Through? Additionally, the sentence needs revision for commas.

Well spoted! Changed: “Throw » Through

Rephrased

Line 238: “The peek-a-boo play is one of earliest forms of play, in which babies and their mothers engage…”. I understand that this sentence comes in accordance with the mother-baby dyad mentioned in the attachment section, however, here, there seems to be a bias by the authors, claiming that only the mother performs this activity. Curiously, in the sentence before, it is recognized that both parents develop interaction with their children. Perhaps clarifying that fathers (or more precisely, the other parent besides the mother; search for inclusive language) also may play peek-a-boo would resolve the issue.

Changed: Mothers » Parents.

Part II

Line 317-319: “The COVID-19 pandemic has contributed to the decrease of child movement due to lockdown and other preventive measures that resulted in a decrease of outdoor areas to play”. Do you have support for this claim? Given that the authors are Portuguese, I suggest reading the Portuguese report REACT-COVID 2.0 (pag. 8), where, in adults, a slight increase in physical activity was detected. Given that physical activity was one of the motives people were allowed to get out of the house during the lockdown, I am not sure that in children, and all countries, this claim is true. Perhaps reducing the assertiveness of the sentence would better reflect some existing disparities.

We sustained our claim based on 2 articles. Although physical activity among adults may have increased, play (and mostly outdoor free play) among children has decreased.

Line 343-345: This is one of the examples where a citation would be needed. Otherwise, it may just be the opinion of the authors, based on logical assumptions.

We add some references that help to sustain our claim.

Lines 370-372: The sentence needs an ending, as it is, it is not understandable.

Rephrased

Lines 386-388: Another example needing a citation given that this does not result from the reflection made in the previous paragraphs.

There has been an increasing of education programs aimed to “teach emotions” among other related topics. On the other hand, there has been a decrease of free play (as showed in the manuscript). It´s hard to get reports (or other manuscripts) that confirm the changes in educational settings.

Line 390: “though”? Perhaps thought.

Well spoted! Changed

Section 2.2.1: Is this an opinion of the authors or a classification represented in the literature? It is not clear.

This classification was made by us, to the purpose of this manuscript. It´s a simplified way to divide the problematics of children (expressed by one of the poles: inhibition or turmoil). This categorization allow us to express the idea, and avoids to go on a large and deep analysis of child psychopathology (which is not the aim of this essay)

Part III

Line 455: To maintain coherence, change “where” to Where.

Nice detail! Changed

Line 636-638: The message in this sentence is very relevant, but has some issues with the English (redundancy and typo)

Rephrased. Hopefully easier to understand without losing its core message.

Lines 747-752. What does “enable channelling homosexual urges into teams,” mean?

Deleted

Conclusion

Line 757: Where was “unconscious motivation” explored in the essay?

There are some references on the manuscript to the unconscious motivation that underpins the engagement in sport activities.

Lines 762-764: Once again, this was not clearly explored throughout the essay. Why use expressions, terms, concepts, and ideas not explored during the paper (e.g., language, space-temporal representations)?

Rephrased.

Reviewer 2 Report

This essay discusses the importance of play and youth sports in child development in depth and detail. I congratulate the authors for the excellent work that in my opinion should be shared not only in academic field, but also at  school and family level.

However, before this study could be considered for publication I suggest that it would be interesting if the importance of play in the natural environment were also highlighted, accompanied by some literature references.

Moreover, I suggest a small integration in paragraph 3.3 Line 499: “Does the engagement in sports foster child development?” Here the authors conclude “that merely practicing sports cannot ensure positive outcomes, but if sports activities are properly conducted and adapted to children developmental stage, then sports can be an outstanding tool to foster positive development of children and youth.” I completely agree with this statement and that is precisely why I think it would be appropriate to spend a few more words on the skills and training that the professional figures called upon to conduct sports activities aimed at children and youngsters should have.

Author Response

Reviewer 2

This essay discusses the importance of play and youth sports in child development in depth and detail. I congratulate the authors for the excellent work that in my opinion should be shared not only in academic field, but also at  school and family level.

Thank you!

However, before this study could be considered for publication I suggest that it would be interesting if the importance of play in the natural environment were also highlighted, accompanied by some literature references.

Good idea! We add the following section:

1.4.5. Importance of Play in Natural Environment

Despite its benefits for child development, the amount of time spent by children in outdoor free play (unstructured activities) has considerably decreased in current societies (Coates & Pimlott, 2019; Kemple et. al., 2016), contributing to more sedentary lifestyles, disconnected from the natural world (Bento & Dias, 2017).

Unstructured outdoor Play has several benefits on child development, health and wellbeing, including physical development, self-regulation and attention, communication and social development, cognitive development and creativity (Kemple et. al., 2016), and imagination and reasoning abilities (Bush, 2004). Outdoor play also contributes to foster physical exercise (and energy bust), to improve social and emotional development (Neto, 2021) and may contribute to establish rapport between parents and their children (Bush, 2004). Due to its benefits, outdoor play has been considered in educational approaches (Coates & Pimlott, 2019; Harris & Bilton, 2019).

References:

Bento, G.& Dias, G. (2017). The importance of outdoor play for young children's healthy development. Porto Biomedical Journal 2(5): 157-160. DOI: 10.1016/j.pbj.2017.03.003

Bush, V. (2004). The Great Outdoors. Essence, 35(3), 196-198.

Coates, J. K. & Pimlott, W. H. (2019). Learning while playing: Children´s Forest School experiences in the UK. British Educational Research Journal, V. 45, (1), 21-40. DOI 10.1002/berj.3491.

Harris, R. & Bilton, H. (2019). Learning about the past: exploring the opportunities and challenges of using an outdoor learning approach. Cambrige Journal of Education, V. 49, (1), 69-91. DOI 10.1080/0305764X.2018.1442416.

Kemple, K. M.; Oh, J, H.; Kenney, E. & Smith-Bonahue, T. (2016). The Power of Outdoor Play and Play in Natural Environments. Childhood Education, 92:6, 446-454, DOI: 10.1080/00094056.2016.1251793

Moreover, I suggest a small integration in paragraph 3.3 Line 499: “Does the engagement in sports foster child development?” Here the authors conclude “that merely practicing sports cannot ensure positive outcomes, but if sports activities are properly conducted and adapted to children developmental stage, then sports can be an outstanding tool to foster positive development of children and youth.” I completely agree with this statement and that is precisely why I think it would be appropriate to spend a few more words on the skills and training that the professional figures called upon to conduct sports activities aimed at children and youngsters should have.

A brief comment was added. We agree this is a very important topic; we did not extend more to avoid falling out of the scope of the manuscript.

Comment added to section 3.3.

Being a competent Children´s sport Coach is high skilled and demanding activity. These coaches must have knowledge about sports training, but also about child development. To become a role model, these coaches must understand the children´s needs, and to focus their attention on his/her best interest at all the time.

Reviewer 3 Report

Comments

Thank you for the opportunity to review the paper titled: Movement, Play, and Games. An essay about youth sports and their benefits for human development. The study purpose of this paper is the importance of play and youth sports on child development. The topic area is understudied, likely due to the lack of the progressive increase of motor repertoire in children’s conceptualization. Thus, this study is timely. The paper presents a current and relevant theme. The theoretical framework is consistent, well used, and provides a reasonable basis for the construction of the methodology. The results obtained are relevant and contribute to their field of study. Overall, this is an interesting study. I have provided detailed feedback in the attached file.

Specific comments are listed below

Introduction  

The paper demonstrates an adequate understanding of the relevant literature in the field and cites an appropriate range of literature sources.

Methods and Results

The results were clearly presented and properly analyzed. The conclusions adequately brought together all the other elements of the study.

Discussion

There are several broad arguments and claims in the manuscript.
Particularly I encourage the authors to highlight the research’s theoretical and practical implications for each hypothesis than simply stating previous research’s suggestions.

Author Response

Reviewer 3

Comments

Thank you for the opportunity to review the paper titled: Movement, Play, and Games. An essay about youth sports and their benefits for human development. The study purpose of this paper is the importance of play and youth sports on child development. The topic area is understudied, likely due to the lack of the progressive increase of motor repertoire in children’s conceptualization. Thus, this study is timely. The paper presents a current and relevant theme. The theoretical framework is consistent, well used, and provides a reasonable basis for the construction of the methodology. The results obtained are relevant and contribute to their field of study. Overall, this is an interesting study. I have provided detailed feedback in the attached file.

Introduction

The paper demonstrates an adequate understanding of the relevant literature in the field and cites an appropriate range of literature sources.

Method and results

The results were clearly presented and properly analyzed. The conclusions adequately brought together all the other elements of the study.

Discussion

There are several broad arguments and claims in the manuscript.
Particularly I encourage the authors to highlight the research’s theoretical and practical implications for each hypothesis than simply stating previous research’s suggestions.

Thank you for your analysis. We believe the comments were properly addressed as the result of the changes made in the last version of the manuscript. If there are any other changes that should be considered, please let us know.  

Round 2

Reviewer 3 Report

Thank you for your efforts for this revision.

Author Response

Reviewer 3

Thank you very much.